# Rural prioritization may increase the impact of COVID-19 vaccines in a representative COVAX AMC country setting due to ongoing internal migration: A modeling study

Prashanth Selvaraj[1]*, Bradley G. Wagner[1], Dennis L. Chao[1], Maïna L'Azou Jackson[2], J. Gabrielle Breugelmans[3], Nicholas Jackson[3], Stewart T. Chang[1]

**1** Institute for Disease Modeling, Bill and Melinda Gates Foundation, Seattle, Washington, United States of America, **2** Coalition for Epidemic Preparedness and Innovations, Oslo, Norway, **3** Coalition for Epidemic Preparedness and Innovations, London, United Kingdom

* prashanth.selvaraj@gatesfoundation.org

**Data Availability Statement:** All code to reproduce the modeling results are available in the following

## Abstract

How COVID-19 vaccine is distributed within low- and middle-income countries has received little attention outside of equity or logistical concerns but may ultimately affect campaign impact in terms of infections, severe cases, or deaths averted. In this study we examined whether subnational (urban-rural) prioritization may affect the cumulative two-year impact on disease transmission and burden of a vaccination campaign using an agent-based model of COVID-19 in a representative COVID-19 Vaccines Global Access (COVAX) Advanced Market Commitment (AMC) setting. We simulated a range of vaccination strategies that differed by urban-rural prioritization, age group prioritization, timing of introduction, and final coverage level. Urban prioritization averted more infections in only a narrow set of scenarios, when internal migration rates were low and vaccination was started by day 30 of an outbreak. Rural prioritization was the optimal strategy for all other scenarios, e.g., with higher internal migration rates or later start dates, due to the presence of a large immunological naive rural population. Among other factors, timing of the vaccination campaign was important to determining maximum impact, and delays as short as 30 days prevented larger campaigns from having the same impact as smaller campaigns that began earlier. The optimal age group for prioritization depended on choice of metric, as prioritizing older adults consistently averted more deaths across all of the scenarios. While guidelines exist for these latter factors, urban-rural allocation is an orthogonal factor that we predict to affect impact and warrants consideration as countries plan the scale-up of their vaccination campaigns.

## Introduction

COVID-19 has presented every country with a challenge to formulate a strategy to protect its population, initially with non-pharmaceutical interventions (NPI)/public health and safety

GitHub repository: https://github.com/
InstituteforDiseaseModeling/selvaraj_spatial_
covid_vax_2021.

**Funding:** The author(s) received no specific
funding for this work.

**Competing interests:** The authors have declared
that no competing interests exist.

measures (PHSM) and more recently vaccination. As of June 2021, the World Health Organization (WHO) has granted Emergency Use Listing to five vaccines [1], though supplies are expected to be limited, particularly in low- and middle-income countries (LMIC). The COVID-19 Vaccines Global Access (COVAX) Advance Market Commitment (AMC) is expected to be the primary mechanism by which most of 92 eligible LMICs procure COVID-19 vaccines, beginning with 3% population coverage for health care workers (HCW) and increasing to 20% population coverage for the elderly and adults with comorbidities [2]. Previous modeling studies have indicated that prioritizing elderly adults for vaccination would yield the largest reductions in mortality [3, 4] given the steep gradient of mortality observed with age in China, Europe, and other countries [5, 6]. Meanwhile, HCW have been prioritized to preserve health system capacity [2].

How countries should allocate vaccines subnationally has received little attention outside of equity or logistical considerations. While the WHO requires countries to submit national deployment and vaccination plans (NDVPs) [7] prior to receiving vaccines from COVAX [8], WHO guidance on developing NDVPs mentions geography only in the context of ensuring equitable access. In particular the guidance encourages countries to give special consideration to "those living in informal settlements or urban slums. . . populations in conflict settings or those affected by humanitarian emergencies, and other hard-to-reach population groups" but does not explore the epidemiological impact of such policy decisions [7]. Likewise, WHO guidance on logistics mentions "remote areas" but only with respect to the ultra-low temperatures required to store certain vaccines and the need for special devices such as thermal shippers with dry ice [9]. These documents highlight the challenges that rural areas (and some urban areas) are expected to face during COVID-19 vaccination campaigns and suggest that the default may be to overlook these areas or distribute vaccine to these areas at a slower pace.

In this study we ask whether urban or rural prioritization of COVID-19 vaccines may also affect the impact of the vaccination campaign on disease burden. Using an agent-based model of an archetypal country from Sub-Saharan African (SSA), which accounts for four of every ten COVAX AMC-eligible countries, we examine the possible impact on transmission and disease burden of campaigns prioritizing urban or rural areas to receive vaccine first, while also accounting for other factors such as age group prioritization, variable dates of vaccine introduction, and final vaccine coverage levels.

## Methods

### Model structure and demographics

COVID-19 transmission dynamics were simulated using EMOD, a software platform for agent/individual-based epidemiological modeling of infectious diseases [10]. EMOD was used to represent an archetypal country in sub-Saharan Africa (SSA) [11] through demographics (age pyramid and urban-rural population distribution), contact structure, and mobility patterns. Demographics were represented by taking the mean of each five year age bin from the population pyramids in the SSA region [12]. The urban-rural distribution of the population was obtained from published UN estimates, and the mean was taken for all countries in SSA [13] (Fig 5). Contact structures for the same set of age bins and countries in urban and rural settings were obtained from Prem et al. [14]. These matrices were available for home, school, workplace, and community settings, and a simple mean was taken across SSA countries. To obtain the total contact matrices, we multiplied the NPI policy effect from the Oxford COVID-19 Government Response Tracker described below and then summed all four settings together. Mobility patterns represented the within-country movement of people as described

**Table 1. Model parameters.**

| Symbol | Parameter | Units | Baseline value | Range | Reference |
|---|---|---|---|---|---|
| $S_{t=0}$ | Initial susceptible prop | Percent | 99.9 | NA | NA |
| $I_{t=0}$ | Initial infected prop | Percent | 0.1 | NA | NA |
| $R_{t=0}$ | Initial removed prop | Percent | 0 | NA | NA |
| $R_0$ | Base reproduction number | New infections / infected indiv | 2.4 | 2.0–2.8 | NA |
| $NPI_t$ | NPI intensity effect | Unitless multiplier | Varies | 0–100 | [15] |
| $\gamma$ | (Avg infec duration)$^{-1}$ | Days$^{-1}$ | $14^{-1}$ | NA | [16] |
| $VE_S$ | Vax effic vs infec | Scalar | 0.95 | {0.95, 0} | [17, 18] |
| $VE_I$ | Vax effic vs trans | Scalar | 0.3 | {0, 0.7} | NA |
| $VE_P$ | Vax effic vs progr | Scalar | 1.0 | {0, 1.0} | [17, 18] |
| $P_{vac}$ | Final vax coverage | Percent | 50 | {20, 50, 80} | [2] |
| $p_{rural}$ | Rural pop | Percent | 60 | NA | [13] |
| $b$ | Baseline migration | Probability / time step | 20 000 dtpmi | 20–200 000 dtpmi | NA |
| $N_{tot}$ | Model agents | Integer | $4 \times 10^5$ | NA | NA |
| $n_{rural}$ | Rural nodes | Integer | 199 | NA | NA |
| $n_{urban}$ | Urban nodes | Integer | 1 | NA | NA |
| $t$ | Time step | Days | 1 | NA | NA |

below. Baseline parameter values are provided in Table 1 except for contact matrices which are provided in S1 Appendix.

Individuals were created at model initialization and assigned (a) an age to match the desired age distribution and (b) a node representing either urban or rural settings to match the desired urban-rural population distribution. In total the model contained 200 nodes with node$_1$ representing an urban setting and node$_{2..200}$ representing rural settings with the same population sizes at initialization (Fig 1). The urban node accounted for 40% of the total population, while the rural nodes accounted for the remaining 60%. This node structure was chosen to represent an SSA country having a capital city serving as the main hub of international traffic and rural areas whose residents travel to the capital city and other rural towns. Bidirectional migration between urban and rural nodes and between rural nodes was represented as a change in the node assignment of individuals. The probability of a change in node assignment was calculated as the product of a base rate and the population sizes of each pair of nodes, i.e., using a gravity-based model (Fig 1). The base migration rate was not known *a priori*, so a range of values were

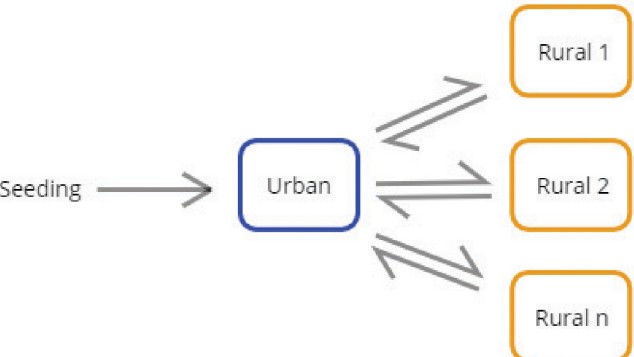

**Fig 1. Model schematic showing sample node and migration.** Infected individuals were seeded in the urban node and infection spread to rural nodes via urban-rural migration.

evaluated spanning several orders of magnitude (Table 1). This rate and other rates in the model were implemented as exponentially distributed waiting times between random events. Seasonal migration or other migration patterns on timescales longer than the incubation or infectious periods of COVID-19 was not explicitly tracked in the model, nor were migration patterns resulting in net changes in the urban or rural population sizes. However, the impact of longer timescale population movement was reflected in the population size differences between urban and rural settings. Other demographic processes such as births, non-disease deaths, and migration to other countries were assumed negligible on the timescale of the model. The model was run for a simulated 2-year (730 day) duration.

## COVID-19 epidemiology in the model

We represented COVID-19 outbreaks where an infected case was introduced into a population that did not have pre-existing immunity (as happened in year 2020) or where a new SARS-CoV-2 variant might be introduced that completely evades existing immunity. Individuals (agents) were assumed to be in one of the following states: susceptible, infected, or removed (recovered or dead). Infected individuals were seeded in the urban node at multiple time points (Table 1) during the first 60 days after which no additional importations were assumed to occur due to suspension of air travel. Susceptible individuals became infected at a rate proportional to a baseline rate of transmission $R_0 I(t)/c_{i,j} \gamma$ where $R_0$ represents the base reproduction number of COVID-19 from literature (Table 1), $I(t)$ the number of infected individuals in the same node, $c_{i,j}$ the mean contact rate of individuals of any two age groups $i$ and $j$, and $\gamma$ the mean infectious duration in the model. $c_{i,j}$ was obtained from the age- and location-dependent contact rates from Prem et al. [14] across settings and countries in SSA. We assumed that infectivity and contact structure were the same for all infected individuals regardless of symptomatic status. Regarding infectivity, multiple studies have shown symptomatic and asymptomatic individuals with PCR-confirmed SARS-CoV-2 infections exhibit similar viral loads as indicated by RT-PCR Ct values [19]. In longitudinal studies individuals who remained asymptomatic throughout their infections, i.e., were not merely pre-symptomatic, exhibited similar viral load profiles to symptomatic individuals [20]. Regarding contact structure, although it is possible infected individuals may self-isolate and curtail contact with others after symptom onset, empirical evidence that this occurs in SSA or LMIC is lacking, particularly in cases that do not require hospitalization [21, 22]. In the model we assumed that 10% of all infected individuals, regardless of symptomatic status, self-isolate reducing their individual contribution to node-level infectiousness via $I(t)$ by 80%. Following infection, we assume individuals transition to the removed state with the sum rate of recovery and death, i.e., the inverse of the total infectious duration. Removed individuals did not return to the susceptible state under the assumption that on the timescale of the model (a) births and deaths were negligible and (b) recovered individuals had perfect immunity that did not wane.

## Country-level events and policies in the model

To represent typical country-level policies to outbreaks, we included a quantification of NPI/PHSM policy strengths implemented in SSA since early 2020. We obtained data from the Oxford COVID-19 Government Response Tracker (OxCGRT) Containment and Health Index which is the average of 14 sub-indexed indicators ranging between 0 and 100 where each sub-index rates a specific type of policy on each day [15]. Dates in OxCGRT were rescaled to the date since first case reported to the WHO ([23], Table 2). To derive a mean value for SSA, we used the mean $NPI_t$ across countries in SSA for each date since respective first case Fig 2. $(1 - NPI_t/100)$ was used as a contact rate multiplier for work and community settings.

**Table 2. Summary of events and policy implementation dates in model.** Calendar day 0 = January 1, 2020.

| Calendar day | Epidemic day ($t$) | Event or policy |
|---|---|---|
| 15 | -60 | Importations begin |
| 75 | 0 | Earliest policies (assumed first reported case) |
| 90 | 15 | Importations stop |
| 105–255 | 30–180 | Vaccinations begin |
| 210 | 135 | Schools reopen |

The school settings was assumed to be closed (i.e., with $NPI_t = 100$) for 135 days after first reported case then reopened with the same contact rate multiplier as work and community settings. The home setting was assumed to have increased contact rates by 25% (i.e., with $NPI_t = -25$). After the last day of available $NPI_t$ data, work, school, and community settings were assumed to be opened to 75% of pre-COVID levels (i.e., with $NPI_t = 25$) to simulate near-complete reopening in most SSA countries. We used the same multiplier $(1 - NPI_t/100)$ on migration between all spatial nodes in the simulation to account for changes in migration because of NPI policies.

## Vaccination in the model

COVID-19 vaccines in the model were represented as protecting individuals by one of the following mechanisms:

1. Acquisition-blocking: preventing susceptible individuals from becoming infected (called efficacy against susceptibility, $VE_S$ in the biostatistical literature [24])

2. Transmission-blocking: preventing shedding of infectious particles after infection (efficacy against infectiousness, $VE_I$, in the literature [24])

3. Disease-blocking: preventing individuals from developing symptomatic disease (efficacy against symptomatic illness, $VE_P$, in the literature [24])

We chose optimistic vaccine characteristics to highlight how despite having the best vaccines available, delayed delivery and not vaccine efficacy within an individual will be more detrimental to overall vaccine impact. Additionally, the effect of vaccines on severe cases and mortality can be approximated as a linear model so a less efficacious vaccine would lead to a commensurately lower effect on averting severe cases and mortality. We assumed that vaccines would be highly efficacious and comparable to the Pfizer/BioNTech or Moderna mRNA vaccines with >90% clinical trial efficacy and effectiveness [17, 18]. Such high efficacy vaccines are anticipated to come into the COVAX supply chain by 2022 [25]. We also made the simplifying assumption that observed efficacy $VE_{obs}$ was entirely attributable to either $VE_S$ or $VE_I$ (see Table 1). That is, for an acquisition-blocking vaccine, we assumed $VE_S = VE_{obs}$ and $VE_P = 1$ resulting in all of the vaccinated individuals being protected from disease progression despite infection. For a disease-blocking vaccine, we assumed $VE_I = VE_{obs}$, $VE_P = 1$, again resulting in all of the vaccinated individuals being protected from disease progression, even if infected. Thus all vaccines, regardless of type, were assumed to prevent severe disease and mortality. In the case of vaccines with lower efficacy, the preventive effect on severe disease and deaths is also assumed to decrease linearly. Finally, as our focus was on factors under the control of country policymakers, we did not explicitly model SARS-CoV-2 variants or their impact on vaccine efficacy. However, we expect that variants would have the net effect of reducing impact similar to reduced coverage levels.

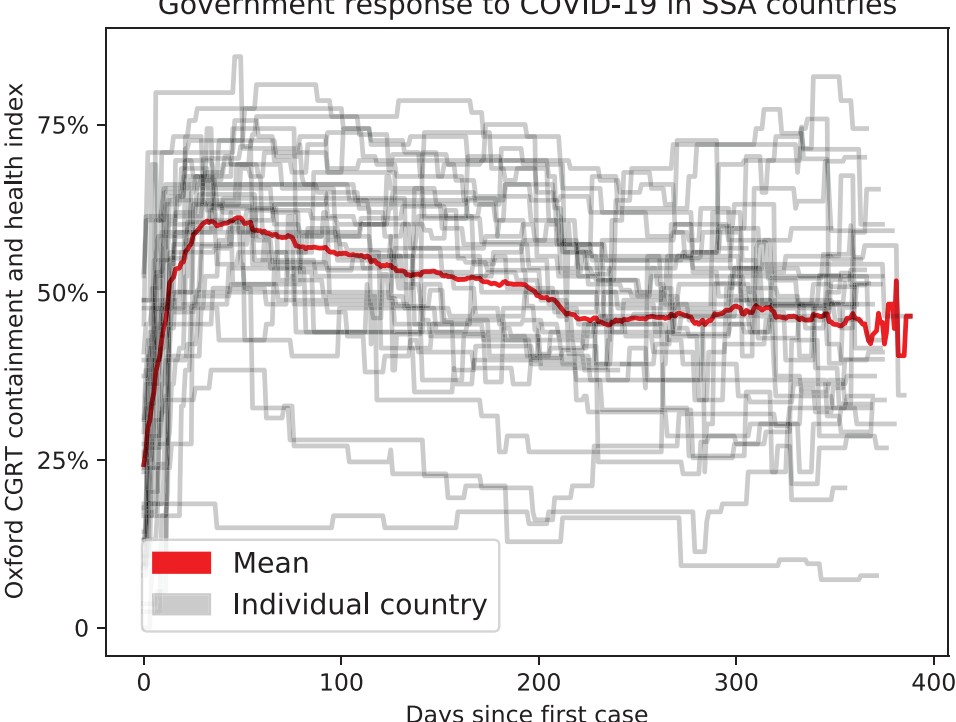

**Fig 2. NPI policies as scored by OxCGRT for SSA countries.**

Vaccine availability followed the COVAX projections of vaccine supply available from GAVI [26]. Briefly these projections assume vaccine will become increasingly available over a period of 12 months until reaching a final vaccination population coverage of $P_{vac}$ = 20% or greater (Table 1).

Different age prioritization schemes were assumed to be available: oldest first (in order: 70+ years, 60–69, 50–59, and 15–49), youngest first (in order: 15–49 years, 50–59, 60–69, and 70+), or random (all ages at the same time) (Fig 3). Likewise, different spatial allocation schemes were assumed viable: urban first, rural first, or random (equal priority to urban and rural areas) (Fig 3). Under random spatial prioritization, vaccine was distributed to urban and rural nodes at a rate proportional to population size, e.g., at 1% coverage, both 1% of the urban population and 1% of the rural population would be covered. In the case of both age and spatial prioritization, age was assumed to take precedence before spatial (urban, rural) setting.

At the individual (agent) level, if an individual was selected to receive a vaccine, each dose was assumed to confer an increasing level of protection until the level of protection specified by $VE_S$, $VE_I$, or $VE_P$ was attained (Fig 4). This profile was chosen to reflect the Moderna scheduling of 28 days between first and second doses [18]. Individual efficacy (regardless of type) was assumed to begin at 0% on day 0 and increase linearly until reaching 80% of final efficacy on day 10. This was maintained until day 28 when a second dose was assumed administered, then efficacy increased linearly until final efficacy was reached on day 35. We assumed that all individuals selected for vaccination would receive both vaccine doses and that individual protection would not wane on the timescale of the model. Vaccination scenarios in the model used the same random number draws as baseline scenarios so that the individual- and population-level characteristics were the same until the vaccination campaign began. Each scenario was run with 40 realizations to derive mean and standard deviation of summary statistics.

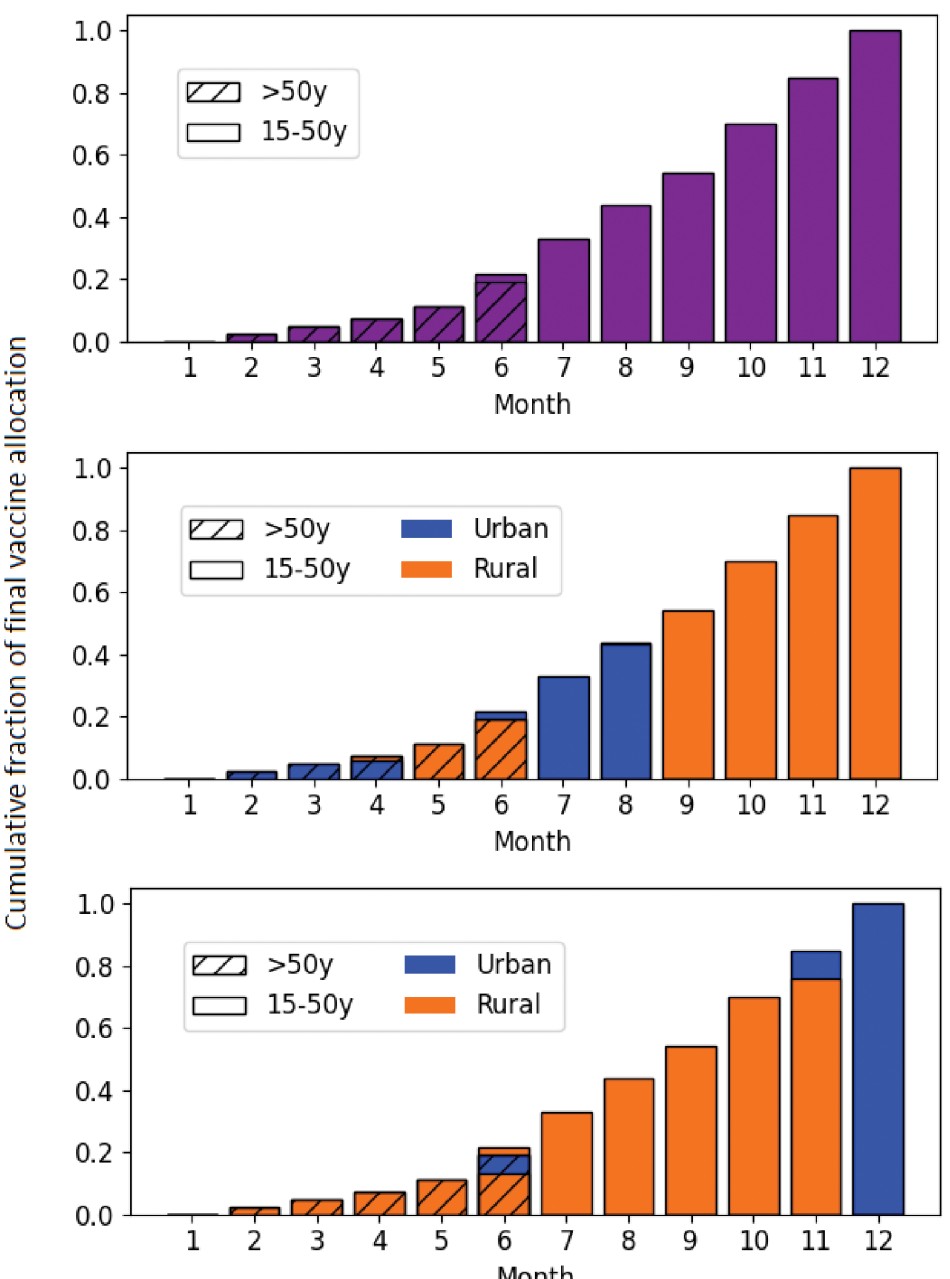

**Fig 3. Ramp-up of protective effect within a vaccinated individual in the model.**

## Parameter ranges for scenarios

We evaluated vaccine distribution scenarios over a range of base transmission intensities represented by $R_0$ and migration rates. In the scenarios presented, $R_0$ varied between 2.0 to 2.8. Migration varied from a scenario where simulated agents make 2000 daily trips per million individuals (dtpmi) under unmitigated (no NPI/PHSM) conditions to a high migration scenario where 200 000 trips are completed each day.

With respect to vaccine distribution, we evaluated different final coverage levels and vaccine distribution start times in addition to different age and spatial prioritization strategies in

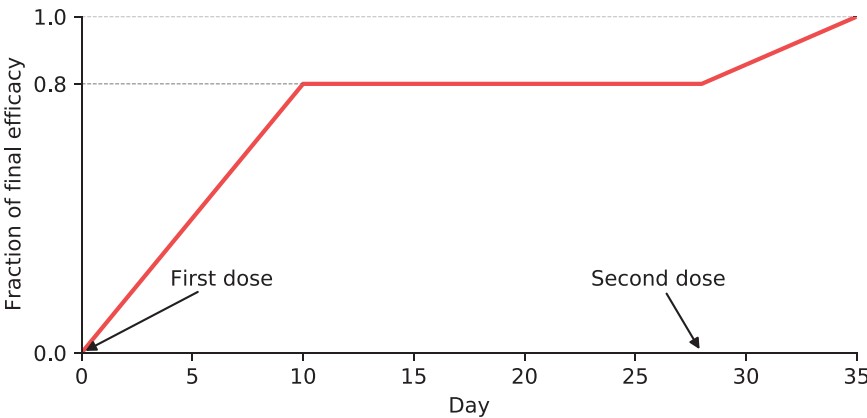

**Fig 4. Vaccine allocation scenarios by spatial prioritization, age group, and month.**

urban and rural areas. Coverage levels were varied from 20%-80% of the total population, while start times were varied between 30–180 days after the first case. While the same start times were used across different transmission scenarios, the number of infected individuals varied depending on transmission intensity preceding the start of vaccine distribution. Vaccine distribution start times could thus be considered a proxy for different levels of population immunity at the start of a vaccination campaign.

Contact structure, age pyramid, and government response tracker data were available for 37 countries in sub-Saharan Africa, and all 37 countries (shown in Fig 5) were used to develop a representative SSA country. A full list of countries is available upon request.

## Results

### Ongoing migration generates multiple peaks in COVID-19 incidence at the country level

To explore possible COVID-19 trajectories in SSA, we created an agent-based model of the spread of COVID-19 in a SSA-like country setting where the population was split between urban and rural areas. The population was assumed to be immunologically naive, and NPI/PHSM policies were simulated as dynamic changes in transmission. Infections were seeded in the urban node and allowed to spread to rural nodes through internal urban-rural migration (Fig 6).

In the baseline scenario, moderate transmission and ongoing migration ($R_0$ = 2.4, migration = 20 000 daily trips per million individuals, dtpmi) resulted in separate incidence peaks in urban and rural areas which aggregated to give two peaks at the country level (Fig 6, middle column, middle row). Cases rose quickly in urban areas and more gradually in rural areas with urban and rural peaks separated by approximately four months. After two years, a majority of the overall population (58% ± 2%) had been infected at some point (Fig 7). A larger fraction of the urban population was ever infected (81% cumulative urban infections vs. 48% cumulative rural infections), but rural infections outnumbered urban infections due to the larger rural population (Fig 7).

Using the baseline scenario as reference, we examined the effects of transmission and migration separately on the epidemic characteristics (Fig 6). Increasing the transmission rate (from $R_0$ = 2.0 to 2.8) while holding migration constant shortened the time between urban and rural peaks (e.g., from 287 to 92 days when migration = 20 000 dtpmi, Fig 6). This also resulted

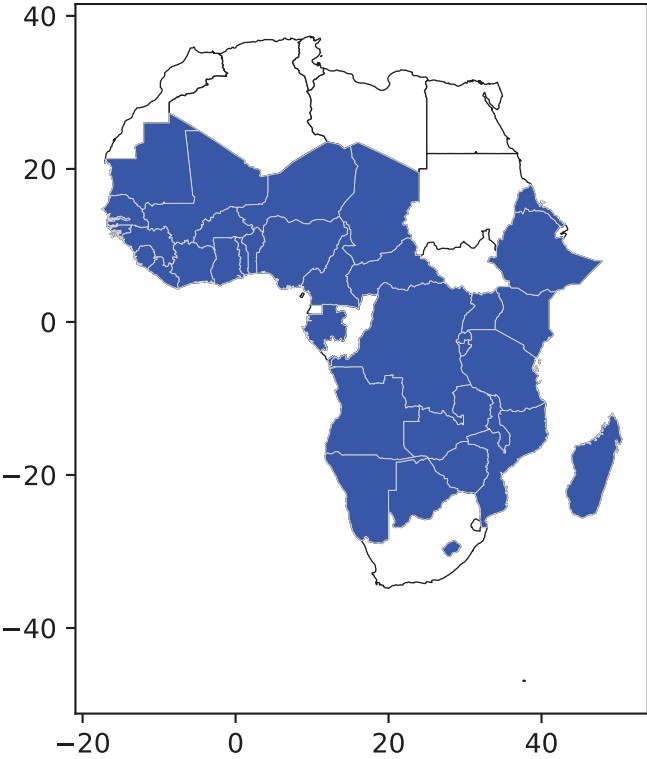

**Fig 5. Sub-Saharan African countries that were aggregated to form a representative country.**

in a proportional increase in cumulative incidence as expected (Fig 7; however, other qualitative features such as the number of peaks at the country level remained unchanged (Fig 6).

By comparison, increasing the internal migration rate while holding transmission constant resulted in qualitative changes to the incidence curve including the number of peaks at the country level (Fig 6). At medium rates of migration (migration = 20 000 dtpmi), separate urban and rural peaks were apparent at the country level with any level of transmission (Fig 6). In this scenario, approximately equal numbers of cases came from urban and rural areas (Fig 7). At low rates of migration (migration = 2000 dtpmi), the outbreak was predominantly urban, with urban areas contributing the majority of cases (Fig 7). Rural incidence failed to increase appreciably but instead contributed to a long tail of declining incidence at the country level (Fig 6). Conversely, at high rates of migration (migration = 200 000 dtpmi), a predominantly rural outbreak was observed, with rural areas contributing the majority of cases (Fig 7). This scenario may represent the urban-to-rural movement of people that occurred despite lockdowns. Incidence increased in both urban and rural areas but continued increasing in rural areas even after starting to decline in urban areas (Fig 6). Therefore, both low and high migration rates tended to produce single-peak outbreaks, while intermediate migration rates resulted in multiple peaks, marking the transition between between predominantly urban and rural outbreaks.

At extreme values of transmission and migration, the model demonstrated two contrasting outbreak scenarios. A more contained, primarily urban outbreak was generated when transmission and migration rates were low ($R_0$ = 2.0 and migration = 2000 dtpmi); cumulative incidence was 62% and 5% in urban and rural areas, respectively, and 22% ± 2% at the country level (Fig 7). A less contained, widespread outbreak was generated when transmission and

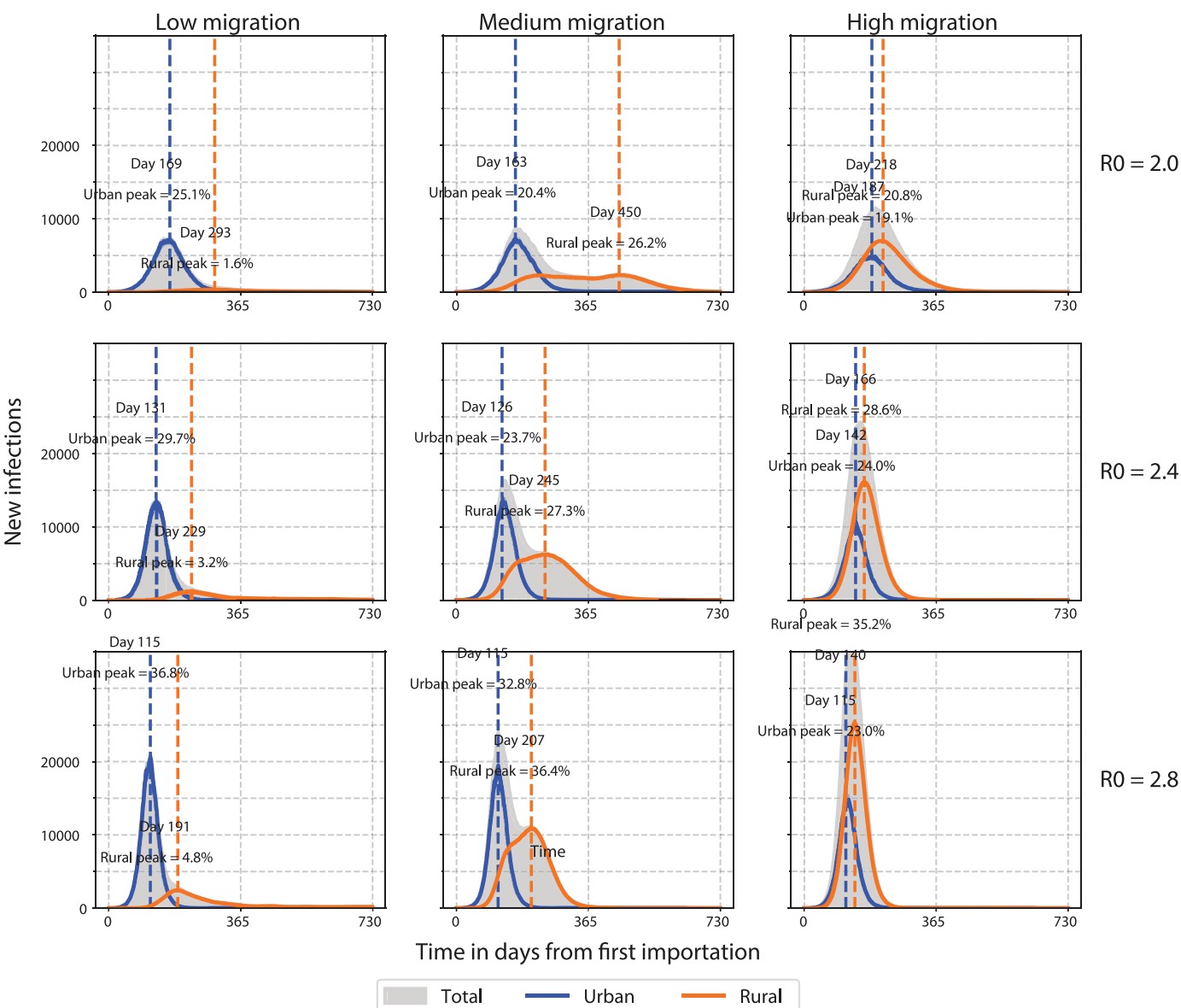

**Fig 6. Daily COVID-19 incidence with varying transmission and migration in an archetypal SSA country setting.** Urban and rural curves show new cases per day out of a population of 400 000 individuals split 40% urban and 60% rural. Peak percentages represent cumulative incidence as a percentage of the respective sub-population on the date when the highest incidence in each sub-population occurs indicated by the dashed vertical line. Time in days since first imported case. Low, medium, and high migration = 2000, 20 000, and 200 000 dtpmi, respectively.

migration rates were high ($R_0$ = 2.8 and migration = 200 000 dtpmi); cumulative incidence was 78% and 63% in urban and rural areas, respectively, and 68% ± 2% at the country level (Fig 7). This range of cumulative incidence was consistent with seroprevalence rates in SSA [27–31]. Finally, our results suggest the urban outbreak is larger per capita than rural outbreaks, which is in line with sero-surveys from a number of countries that are representative of the broader population [32–34].

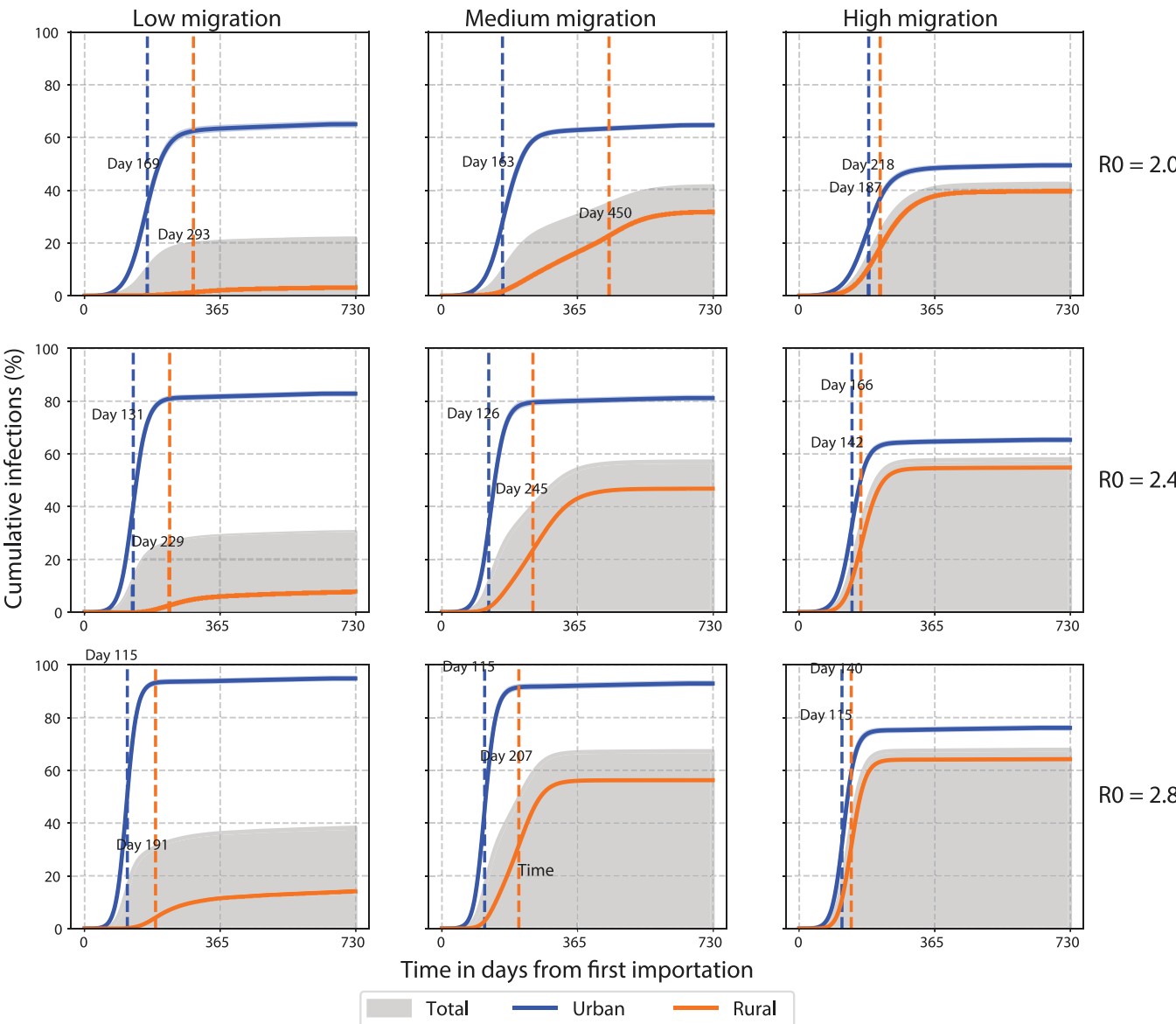

**Fig 7. Cumulative COVID-19 incidence with varying transmission and migration in an archetypal SSA country.** Urban and rural curves show percentage ever infected out of a population of 400 000 individuals split 40% urban and 60% rural. Peak percentages represent cumulative incidence as a percentage of the respective sub-population on the date when the highest incidence in each sub-population occurs indicated by the dashed vertical line. Time in days since first imported case. Low, medium, and high migration = 2000, 20 000, and 200 000 dtpmi, respectively.

## Increasing coverage has a limited capacity to compensate for delays in COVID-19 vaccine introduction

To examine the effect of COVID-19 vaccination policies on population-level impact, we simulated roll-out strategies that differed in several respects: age prioritization (i.e., older or younger adults first), timing (start day) of vaccine introduction, final vaccine coverage level, and spatial prioritization (i.e., urban or rural areas first). Vaccines were assumed to be either acquisition-blocking (preventing most infection after exposure) or disease-blocking (allowing infection and limited transmission but preventing most disease). Impact was evaluated on the basis

of cumulative infections, severe cases, and deaths averted after two years compared to a counterfactual baseline scenario of moderate transmission and ongoing migration ($R_0 = 2.4$, migration = 20 000 dtpmi).

We first asked what effect age prioritization and timing of introduction might have on vaccination impact (Fig 8). To simplify the analysis, we fixed vaccine coverage at 50% and distributed vaccine to both urban and rural areas with equal priority. We found that the optimal age prioritization strategy depended on the choice of impact metric. Prioritizing younger adults to receive an acquisition-blocking vaccine averted more infections than prioritizing older adults or random distribution by age (e.g., 15%±3% versus 13%±2% and 7%±2%, respectively, when introduced at day 60, Fig 8, left column, top row), consistent with the large proportion of younger adults in SSA populations [13] and degree of contact that occurs within this age group [14]. With a disease-blocking vaccine, prioritizing younger adults continued to be more effective in averting infections than prioritizing older adults, but the relative differences were reduced (Fig 8, left column, bottom row).

By comparison, prioritizing older adults was consistently more effective at averting severe cases or deaths than prioritizing younger adults or random distribution by age. For an acquisition-blocking vaccine, prioritizing older adults averted approximately twice as many deaths as prioritizing younger adults, regardless of date of introduction (Fig 8, right column, top row). These differences were greater in magnitude for a disease-blocking vaccine (Fig 8, right column, bottom row). Regardless of impact metrics, delays in vaccine introduction reduced but did not negate the absolute differences in impact between age prioritization strategies (Fig 8).

We then asked whether higher coverage levels could compensate for a later date of introduction. In this case, we assumed priority would go to older adults based on the deaths averted as a metric (Fig 9) and WHO fair allocation guidelines [2]. Final coverage levels were varied

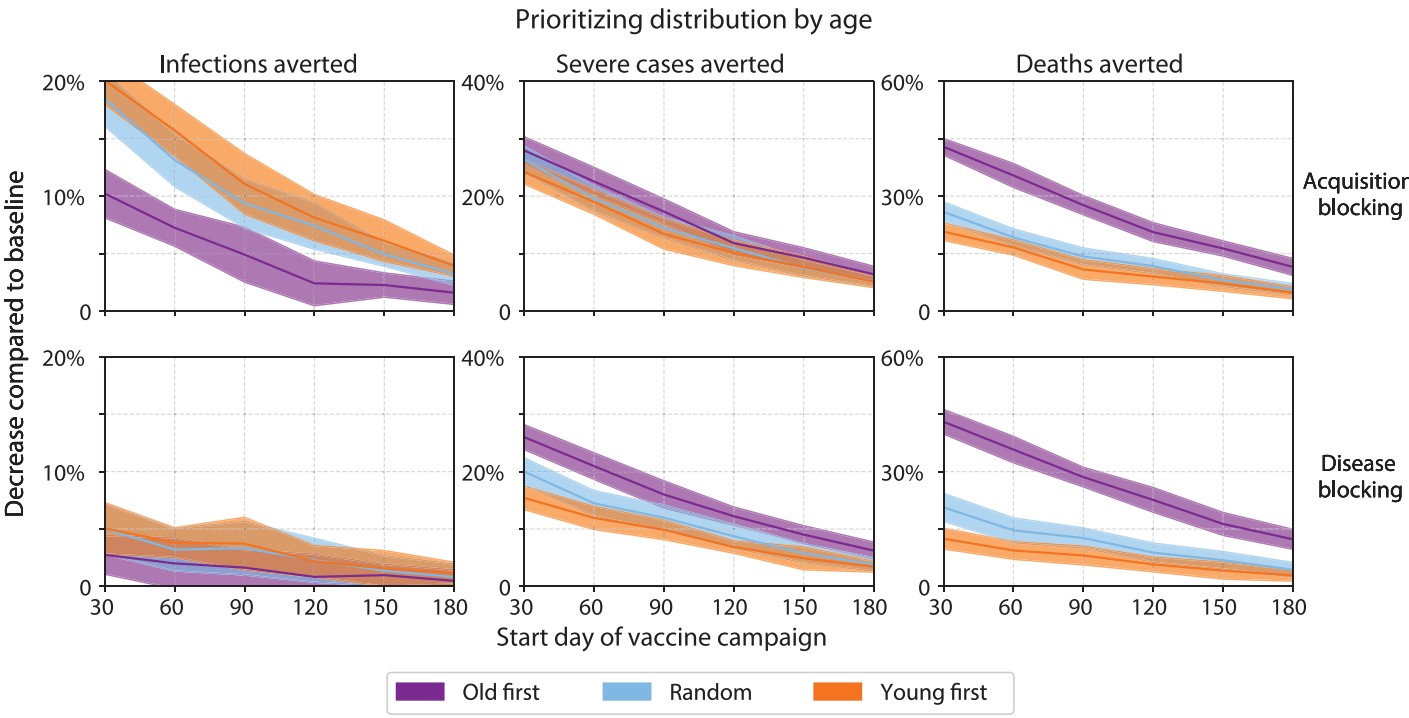

**Fig 8. Vaccination impact by age prioritization and timing of vaccine introduction.** Age groups for prioritization were as follows: "old first" = beginning with adults age 70 years or older; "young first" = beginning with adults age 15–49 years; "random" = without prioritization (i.e., all ages have equal priority).

**Fig 9. Vaccination campaign impacts by final coverage level and timing of vaccine introduction.** Percentage reductions are shown for the respective cumulative indicator (infections, severe cases, or deaths) with respect to the baseline scenario without intervention.

between 20% and 80% representing COVAX targets [2] and an optimistic scenario with additional vaccine procurement, respectively. Regardless of metric (infections, severe cases, or deaths averted) or start day of vaccination (30–180 days after first case), increasing vaccination coverage also increased the impact (Fig 9).

The maximum possible impact for any given coverage level diminished rapidly with delays in the start of vaccination (Fig 9). An optimistic campaign with 80% coverage of an acquisition-blocking vaccine averted 16% of infections when introduced on day 30 but only 2% of infections when introduced on day 180. The impact of vaccination at other coverage levels was similarly diminished by delays in introduction (Fig 9). As a result, higher coverage levels were necessary for a campaign started later to have the same impact as a campaign started earlier with lower coverage levels. In certain cases, the gain from starting vaccination earlier could not be matched by delayed vaccination, even with very high final coverage levels. For example, to avert the same number of deaths (32% of baseline deaths) as a campaign started on day 30 with 30% coverage, a campaign started on day 90 would need to attain 80% coverage (Fig 9, right column, top row). However, campaigns started on day 30 with ≥60% coverage resulted in levels of impact (≥44% of baseline deaths averted) that could not be achieved by campaigns started later, regardless of coverage level (examined up to 80% coverage) (Fig 9, right column, top row).

Trends with respect to age prioritization, timing of introduction, and final coverage levels were robust to choice of baseline scenarios, including changing the level of transmission or ongoing migration (S1 Appendix).

## Rural prioritization optimizes vaccination in the presence of ongoing migration

Finally we asked to what extent spatial (urban or rural) prioritization could affect the impact of a COVID-19 vaccination campaign in an archetypal SSA setting. We again assumed that

priority would be given to older adults but that within each age tier, those in the prioritized spatial setting would be vaccinated first. For example, for rural prioritization, we assumed vaccine would go to rural individuals 70+ years, then urban 70+ years, rural 60–69 years, and so on. We also varied final coverage levels (20%, 50%, or 80%), start dates of introduction (30, 90, or 180 days), and type of vaccine (acquisition- or disease-blocking). For our counterfactual, we assumed a baseline of moderate transmission and ongoing migration ($R_0$ = 2.4, migration = 20 000 dtpmi) or the extremes of a more confined or less confined outbreak ($R_0$ = 2.0, migration = 2000 dtpmi or $R_0$ = 2.8, migration = 200 000 trips dtpmi, respectively) (cf. Figs 6 and 7).

For the majority of vaccination scenarios, we found that rural prioritization would avert more infections, severe cases, and deaths than urban prioritization (Fig 10). Urban prioritization achieved greater impact for a small set of conditions: when the outbreak was largely confined to urban areas (with low transmission and low migration) and vaccination was started early in the outbreak (by day 30) (Fig 10, left column). For all other scenarios, including those with higher transmission or migration, or campaigns started at later dates, rural prioritization resulted in greater impact (Fig 10, center and right columns). These differences were most pronounced when impact was measured by infections averted. For other metrics such as severe cases (Fig 10, rows 2 and 5), the differences between urban and rural prioritization tended to be modest but still significant, e.g., with medium transmission and migration ($R_0$ = 2.4, migration = 20 000 dtpmi, medium migration). This was attributable to the prolonged spread of the epidemic in such scenarios (Fig 6) as well as our assumption that age groups would be given priority over spatial distribution such that in all scenarios in Fig 10, older age groups were vaccinated first.

The acquisition-blocking vaccine had a higher impact on transmission than the disease-blocking vaccine, particularly when impact was measured by infections averted. However, differences in impact between the two vaccine types were strongly dependent on transmission trajectories over time, coverage, and the start time of vaccine roll-out. In the low migration and low transmission scenario ($R_0$ = 2.0, migration = 2000 dtpmi), the two vaccines were largely similar. For example, for campaigns prioritizing urban areas started on day 30, both vaccines had negligible effects on infections averted with 20% final coverage. These differences increased to 7% of infections (10% and 3% with acquisition- and disease-blocking, respectively) with 80% final coverage (Fig 10, left column) and decreased with delays in the start date of vaccine roll-out. When transmission and migration were both higher ($R_0$ = 2.4, migration = 20 000 dtpmi, medium migration), the epidemic persisted longer (Fig 6), and larger magnitude differences between vaccine types were observed. For campaigns prioritizing rural areas started on day 30, the difference between vaccine types was 3% of infections with 20% final coverage and 15% of infections with 80% final coverage (Fig 10, middle column). These modest absolute differences in impact were attributable to the slow roll-out of vaccine over a 12-month period, particularly during the first 9 months when <50% of the vaccine will have been distributed [26].

The differences between acquisition-blocking and disease-blocking vaccines were reduced when impact was measured by severe cases and deaths (Fig 10). The acquisition-blocking vaccine resulted in 2–4% more severe cases averted than the disease-blocking vaccine across transmission scenarios due to the reduction in transmission (Fig 10, rows 2 and 5). However, the difference between vaccine types was negligible for deaths averted (Fig 10, rows 3 and 6). This was attributable to our assumption that both vaccines provide protection against severe disease and deaths and that age groups would be given priority over spatial distribution.

In all three transmission scenarios in Fig 10 as well as the remaining scenarios (S1 Appendix), increased NPI/PHSM that also reduced migration maximized the impact of vaccination campaigns.

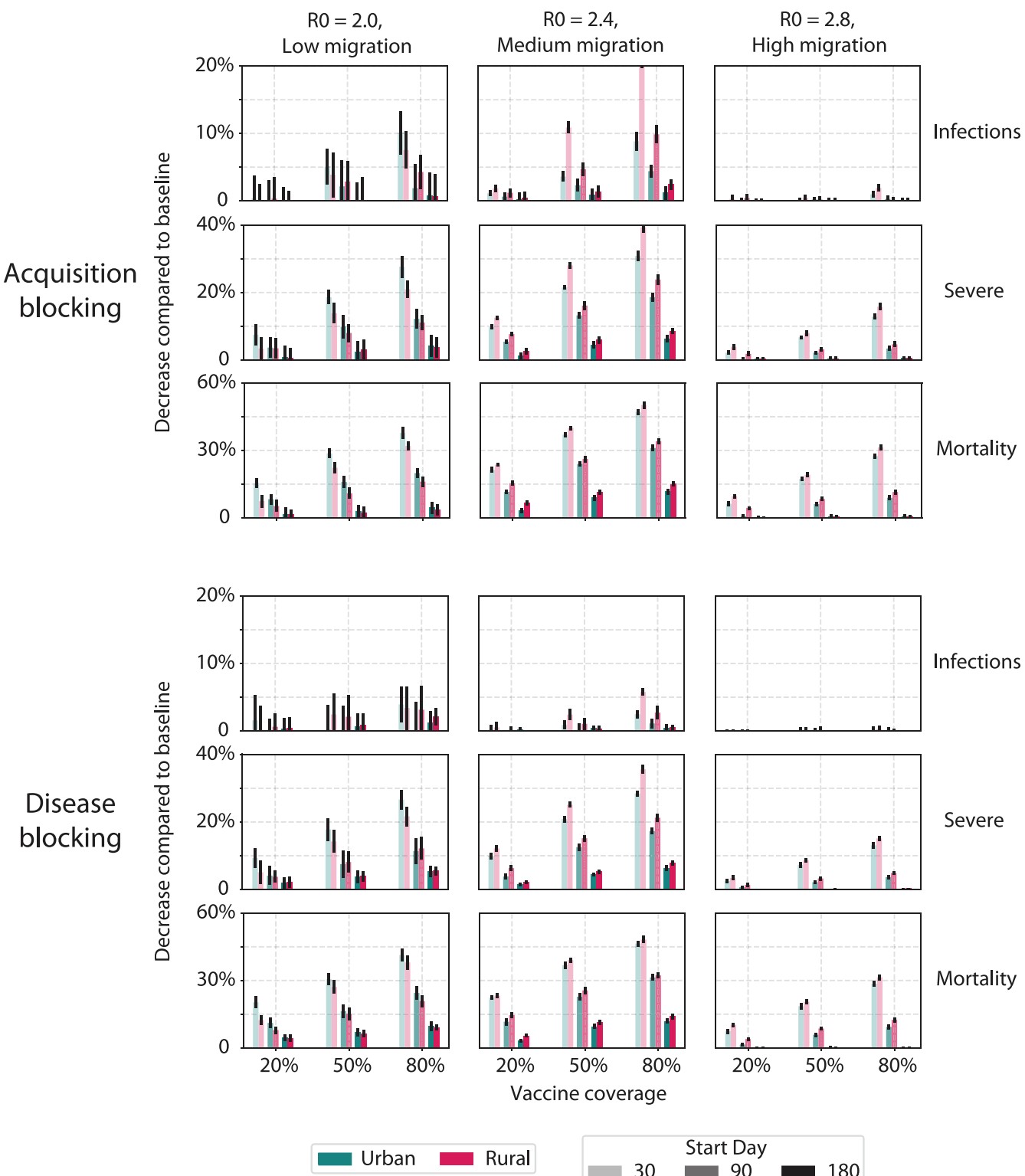

**Fig 10. Vaccination campaign impact on infections, severe cases and deaths by spatial (urban-rural) prioritization.** Bars indicate mean reduction in respective cumulative indicator, and error bars indicate standard error.

## Discussion

By applying an agent-based model with discrete individuals, we found that ongoing urban-rural migration has the potential to extend COVID-19 outbreaks in a SSA-like setting. With either low or high internal migration rates, our model predicted that a single incidence peak at the country level would follow introduction, predominated by cases in urban and rural areas, respectively. However, with intermediate migration rates, our model predicted that an initial predominantly urban outbreak would be followed by a second rural outbreak, resulting in two peaks and extending the duration of the outbreak.

Each of these migration scenarios was in turn associated with a different optimal vaccination strategy. When migration rates were low (whether due to geography, NPI/PHSM, or other causes) and if the vaccine could be rolled out quickly, the optimal strategy was to target urban areas, preventing the predominantly urban outbreak. However, with higher rates of urban-rural migration (as observed in other LMICs after cases appeared [35]), targeting rural areas resulted in greater impact and was the optimal strategy, consistent with the largely rural population in SSA. Delays in vaccination also resulted in rural prioritization becoming the optimal strategy, as urban areas experienced outbreaks faster than rural areas and a larger proportion of the population resided in rural areas.

The differences in impact between urban and rural prioritization strategies warrant consideration as countries begin vaccinating their populations in larger numbers. We are not aware of any NDVPs (such as made available on the WHO COVID-19 Partners Platform [36]) that distinguish between urban and rural areas, though a few countries such as Mexico are reportedly prioritizing rural areas [37]. While WHO guidance on subnational allocation has been limited to equity or logistical considerations, possible effects on impact were not explored [7–9]. If resources are not intentionally directed to rural areas, urban allocation is likely to be the default strategy throughout SSA and other LMIC. For example, Ameyaw et al. examined Demographic and Health Survey data from 2010–2018 and found that children in urban areas were fully vaccinated at higher rates than their rural counterparts (by 53% to 41%, respectively) [38]. While we do not consider other mitigating factors such as the ease of distribution in urban areas, the question of whether to vaccinate a larger, immunologically naive rural population or a smaller, previously exposed urban population encapsulates what we expect is a dilemma that countries may be facing in the near future. In this study we found that vaccinating rural areas may have benefits beyond equity.

Spatial prioritization complemented other strategies such as prioritizing older adults, as recommended by official WHO guidance [2]. Prioritizing younger adults with a vaccine effective against acquisition reduced the overall number of infections, but prioritizing older adults more greatly reduced severe cases and deaths. The inability of prioritizing younger age groups to prevent as many severe cases and deaths as prioritizing older age groups was ultimately due to the anticipated slow roll-out of vaccine. This in turn reflects current projections which foresee a 12-month roll-out to reach 20% coverage that is back-loaded; approximately 50% of the available doses is expected to come in the last three months [26]. This pace, even assuming coverage rates >20%, makes it difficult to achieve herd immunity fast enough to prevent infections, severe cases, and deaths in older adults without direct targeting. These results are consistent with other recent models on COVID-19 vaccine impact [3, 4, 39, 40].

Our model also supports the position that countries should begin their vaccination campaigns as soon as possible. Campaigns that started later had a limited capacity to make up for the delay in vaccination, even when higher coverage levels were assumed. In many scenarios, larger campaigns that started later resulted in more deaths than smaller campaigns that started

earlier. This supports the urgency to direct vaccine to SSA as quickly as possible, a position espoused by COVAX [41], the African Union [42], and civil society [43].

One question that we did not address directly is which scenario currently best fits each country in SSA: which have high or low ongoing internal migration, or high or low transmission? Because both of these parameters are abstracted in the model, we propose that a careful epidemiological assessment would be needed at a country level to test if ongoing or seasonal migration were having an effect, e.g., as a high cumulative incidence in rural areas compared to urban areas. This might be accomplished through serosurveys as in Niger State in Nigeria [29] or through genomic surveillance to track variants as in South Africa [44], Nigeria [45], and Kenya [46]. Both studies showed that infections spread from urban centers to rural areas and that rural areas were slower to see increases in incidence, consistent with a degree of restricted migration. These and other countries in SSA may still have large immunologically naive populations in rural areas.

Our model was informed by a mix of historical data but plausibly represents a future scenario. For example, while almost all SSA countries saw initial outbreaks during the year 2020, the example of Manaus, Brazil demonstrates how new variants may evade even high levels of existing immunity [47]. In such situations, future outbreaks may continue to resemble new outbreaks, leading to successive almost-memoryless incidence peaks. In addition, the vaccine roll-out in SSA has been limited thus far, making our assumption reasonable that future vaccinations will essentially be starting anew. As of June 2021, the Africa CDC reported that less than 2% of member country populations received at least one dose of any vaccine and less than 0.5% have been fully vaccinated [48]. Therefore, variants and low existing vaccine-induced immunity suggest our model and other outbreak-type models will continue to have relevance.

Our study had several limitations. As an abstraction of a SSA country setting, our model represents an average in many respects: demographics (ages, contact patterns, urban-rural localization), COVID-19 response (as tracked by Oxford CGRT [15]), and a generic urban-seeded outbreak. This abstraction represents a trade-off that allowed us to focus on outbreak scenarios that may be applicable to many countries, though not precisely calibrated to any particular country. Our model also had several parameters that were abstractions of physical processes and not precisely matched to data. For example, internal migration did not correspond to a particular indicator such as mobile phone-based movement but represents an aggregate of factors contributing to short-term migration, as distinct from seasonal migration [49]. We also assumed that changes in behavior were governed by government NPI policies, while in reality population behavior could change before or after policies are implemented [50]. We also made several simplifying assumptions on COVID-19 immunity and vaccines such as previous infection leading to perfect immunity and optimistic vaccine characteristics such as those based on mRNA vaccines [17, 18], as well as no opportunity for reinfection. We also did not explicitly account for any particular SARS-CoV-2 variants. However, by spanning a range of transmission rates, we accounted for increased infectiousness expected of new variants. In this case, our selected vaccine efficacy rates may represent best-case scenarios, with variants further reducing the impact on infections, severe cases, and deaths.

In sum, our model supports the position that countries should consider spatial prioritization among other factors when planning how to distribute COVID-19 vaccine, particularly those in SSA where vaccine supplies are expected to be limited. These countries have an ongoing opportunity to adapt their strategy and if necessary set up an infrastructure that allows vaccine to be prioritized to maximize impact.

## Supporting information

**S1 Appendix. Appendix containing additional figures.**
(PDF)

## Acknowledgments

The authors would like to thank Jonathan Bloedow and Christopher Lorton from the Bill and Melinda Gates Foundation for software support and Bob Small and Amol Chaudhari from the Coalition for Epidemic Preparedness Innovations and the WHO SAGE Working Group on Covid-19 Vaccines for informative discussions. This publication is based on research conducted by the Institute for Disease Modeling at the Bill and Melinda Gates Foundation.

## Author Contributions

**Conceptualization:** Prashanth Selvaraj, Bradley G. Wagner, Dennis L. Chao, Maïna L'Azou Jackson, J. Gabrielle Breugelmans, Nicholas Jackson, Stewart T. Chang.

**Formal analysis:** Prashanth Selvaraj.

**Investigation:** Prashanth Selvaraj.

**Methodology:** Prashanth Selvaraj, Bradley G. Wagner, Dennis L. Chao, Stewart T. Chang.

**Project administration:** Prashanth Selvaraj.

**Software:** Prashanth Selvaraj.

**Visualization:** Prashanth Selvaraj, Bradley G. Wagner, Stewart T. Chang.

**Writing – original draft:** Prashanth Selvaraj, Dennis L. Chao, Stewart T. Chang.

**Writing – review & editing:** Prashanth Selvaraj, Bradley G. Wagner, Dennis L. Chao, Maïna L'Azou Jackson, J. Gabrielle Breugelmans, Nicholas Jackson, Stewart T. Chang.

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
