## [Decision Letter · Decision Letter 0]

7 Sep 2021

PGPH-D-21-00314

Rural prioritization may increase the impact of COVID-19 vaccines in a representative COVAX AMC country setting due to ongoing internal migration: A modeling study

Dear Dr. Selvaraj,

Thank you for submitting your manuscript to PLOS Global Public Health. After careful consideration, we feel that it has merit but does not fully meet PLOS Global Public Health’s publication criteria as it currently stands. Therefore, we invite you to submit a revised version of the manuscript that addresses the points raised during the review process.

We look forward to receiving your revised manuscript.

Kind regards,

Claudia Cristina de Aguiar Pereira, Ph.D.

Academic Editor

Journal Requirements:

Additional Editor Comments (if provided):

Reviewers' comments:

Reviewer's Responses to Questions

**Comments to the Author**

1. Does this manuscript meet PLOS Global Public Health’s publication criteria? Is the manuscript technically sound, and do the data support the conclusions? The manuscript must describe methodologically and ethically rigorous research with conclusions that are appropriately drawn based on the data presented.

Reviewer #1: Yes

Reviewer #2: Partly

2. Has the statistical analysis been performed appropriately and rigorously?

Reviewer #1: Yes

Reviewer #2: I don't know

3. Have the authors made all data underlying the findings in their manuscript fully available (please refer to the Data Availability Statement at the start of the manuscript PDF file)?

Reviewer #1: Yes

Reviewer #2: Yes

4. Is the manuscript presented in an intelligible fashion and written in standard English?

Reviewer #1: Yes

Reviewer #2: Yes

5. Review Comments to the Author

Reviewer #1: Dear Authors,

I have reviewed the paper very carefully. This paper is very well-written and lots of novel analysis are there. Your manuscript proposed a lot of information on the dynamics of COVID-19. Just few minor changes are required. I suggest the followings:

1) Some typo mistakes are present in the manuscript. Please look into the paper carefully and correct them.

2) Cite the following papers:

Kumar, P., Erturk, V. S., & Murillo-Arcila, M. (2021). A new fractional mathematical modelling of COVID-19 with the availability of vaccine. Results in Physics, 24, 104213.

Kumar, P., Erturk, V. S., Murillo-Arcila, M., Banerjee, R., & Manickam, A. (2021). A case study of 2019-nCOV cases in Argentina with the real data based on daily cases from March 03, 2020 to March 29, 2021 using classical and fractional derivatives. Advances in Difference Equations, 2021(1), 1-21.

Thank you.

Reviewer #2: The paper considers the effect of prioritizing urban or rural communities in the distribution of COVID-19 vaccines in averting infection within the Sub-Sahara African (SSA) region. The study uses an agent-based epidemiological model as implemented in Epidemiological MODeling software (EMOD).

The paper is well designed and structured. The simulations are well thought, and good number of scenarios are analysed before coming to conclusion. The analysis and conclusion of the paper are interesting and practical.

However, the following are some of the shortcomings identified by this reviewer.

(1) In adapting the software EMOD, the authors did not indicate that they have considered and implemented in the model the difference in the epidemiological behaviour of the Symptomatic and Asymptomatically infectious individuals. It is natural to assume that the contact rates between individuals with symptomatic and asymptomatic persons vary significantly. It is reported in many of the epidemiological literature that most young individuals are observed not to show the symptoms of the disease. If this is true, it might significantly affect the dynamics of the disease in the society where a large proportion of its population is younger in age, like in the case of SSA region.

(2) When an NPI (specifically a lockdown rule) is introduced in many of the countries in the region, it has been observed that a large proportion of people travelled from the urban areas into rural areas before the rule is imposed. The model seems to not consider this fact into account.

(3) In the consideration of immigration, the direction of movement is not clearly indicated in the manuscript. If the direction is from the rural areas to urban ones, the direction is not always the same. Specially, the movement of the younger individuals depends on the time of the year and the need for the workforce in the economic activities. It is not clear to the reviewer how this situation is handled in the simulation.

(4) The final recommendation is based on the baseline scenarios described on pages 14 – 18. Except for the low migration scenario, all of them imply that the total number of infections in rural areas outnumber those in the urban areas. However, as a person experiencing the situation from within the region this reviewer can witness that, this does not reflect the reality on the ground. The infection is still very rare in most of the rural areas in the region. The authors need to check their assumptions to the data on the ground.

On the other hand, authors compared their results on page 18 (line numbers 258 – 262) as consistent with the one indicated in Ref. 22 – Ref. 26. But the samples in these references can only represent a high-risk segment of the population and can not be compared to the situation in the rural setting of the region.

Considering this issue might help experts to re-consider their modelling and data collection approach for the rural areas of the SSA region. Otherwise, the recommended policy directions will not lead to the desired outcome.

(5) A minor but important comment: The vaccine type that is used in the study to simulate the impact is assumed to be highly efficacious and comparable to the Pfizer or Moderna mRNA vaccines (as indicated on page 9, line 140). However, it is known that the vaccine types that are distributed via COVAX are less efficacious than the one assumed in the model system. It is not clear why authors used such an assumption.

6. PLOS authors have the option to publish the peer review history of their article (what does this mean?). If published, this will include your full peer review and any attached files.

**Do you want your identity to be public for this peer review?** For information about this choice, including consent withdrawal, please see our Privacy Policy.

Reviewer #1: **Yes: **Pushpendra Kumar (Institute: NIT Puducherry)

Reviewer #2: No

---

## [Decision Letter · Decision Letter 1]

15 Nov 2021

Rural prioritization may increase the impact of COVID-19 vaccines in a representative COVAX AMC country setting due to ongoing internal migration: A modeling study

PGPH-D-21-00314R1

Dear Dr. Selvaraj,

We're pleased to inform you that your manuscript has been judged scientifically suitable for publication and will be formally accepted for publication once it meets all outstanding technical requirements.

Within one week, you'll receive an e-mail detailing the required amendments. When these have been addressed, you'll receive a formal acceptance letter and your manuscript will be scheduled for publication.

An invoice for payment will follow shortly after the formal acceptance. To ensure an efficient process, please log into Editorial Manager at https://www.editorialmanager.com/pgph/ click the 'Update My Information' link at the top of the page, and double check that your user information is up-to-date. If you have any billing related questions, please contact our Author Billing department directly at authorbilling@plos.org.

Kind regards,

Claudia Cristina de Aguiar Pereira, Ph.D.

Academic Editor

Additional Editor Comments (optional):

Reviewers' comments:

Reviewer's Responses to Questions

**Comments to the Author**

1. If the authors have adequately addressed your comments raised in a previous round of review and you feel that this manuscript is now acceptable for publication, you may indicate that here to bypass the “Comments to the Author” section, enter your conflict of interest statement in the “Confidential to Editor” section, and submit your "Accept" recommendation.

Reviewer #1: All comments have been addressed

Reviewer #2: All comments have been addressed

2. Does this manuscript meet PLOS Global Public Health’s publication criteria? Is the manuscript technically sound, and do the data support the conclusions? The manuscript must describe methodologically and ethically rigorous research with conclusions that are appropriately drawn based on the data presented.

Reviewer #1: Yes

Reviewer #2: Yes

3. Has the statistical analysis been performed appropriately and rigorously?

Reviewer #1: Yes

Reviewer #2: N/A

4. Have the authors made all data underlying the findings in their manuscript fully available (please refer to the Data Availability Statement at the start of the manuscript PDF file)?

Reviewer #1: Yes

Reviewer #2: Yes

5. Is the manuscript presented in an intelligible fashion and written in standard English?

Reviewer #1: Yes

Reviewer #2: Yes

6. Review Comments to the Author

Reviewer #1: Manuscript is suitable for the publication. I accept it.

Reviewer #2: Authors have adequately addressed the previously indicated comments. However, though it can not be considered as a limitation to the manuscript, this reviewer still have two (minor) concerns.

(1) For COVID-19 case, arguably the main variation between the symptomatic and asymptomatic individuals is in their possible contact rates with susceptible individuals. Treating both groups a if they produce the same force of infection is one of the week side of using the EMOD system as it is for the simulation of COVID-19 epidemiological features. Introducing a new cohort of asymptomatic individuals in the model and modifying the EMOD system would have made it acceptable.

(2) Since the study focusses on the classification of rural and urban areas, it would be much better if authors give how they define the classification. For example, some consider regional towns/cities (usually with a population of more than 50,000) as urban and others take them as rural.

7. PLOS authors have the option to publish the peer review history of their article (what does this mean?). If published, this will include your full peer review and any attached files.

**Do you want your identity to be public for this peer review?** For information about this choice, including consent withdrawal, please see our Privacy Policy.

Reviewer #1: No

Reviewer #2: No
